# Overview, Diagnosis, and Perioperative Systemic Therapy of Upper Tract Urothelial Carcinoma

**DOI:** 10.3390/cancers15194813

**Published:** 2023-09-30

**Authors:** Adam Kolawa, Anishka D’Souza, Varsha Tulpule

**Affiliations:** 1IRD 620, Department of Internal Medicine, University of Southern California, 2020 Zonal Avenue, Los Angeles, CA 90033, USA; adam.kolawa@med.usc.edu; 2USC Norris Comprehensive Cancer Center, 1441 Eastlake Ave, Los Angeles, CA 90033, USA; varsha.tulpule@med.usc.edu

**Keywords:** upper tract urothelial, perioperative, immunotherapy

## Abstract

**Simple Summary:**

As upper tract urothelial carcinoma is a relatively rare disease, much of clinical practice has been extrapolated from urothelial carcinoma data. Here we summarize data, current guidelines, and future directions in the management of upper tract urothelial carcinoma with a particular focus on systemic therapy.

**Abstract:**

Upper tract urothelial carcinoma comprises 5–10% of all urothelial carcinoma cases. This disease tends to have a more aggressive course than its lower urinary tract counterpart, with 60% of patients presenting with invasive disease and 30% of patients presenting with metastatic disease at diagnosis. The diagnostic workup of UTUC involves imaging with CT urogram, urine cytology, and direct visualization and biopsy of suspected lesions via ureteroscopy. Standard treatment of high-grade UTUC involves radical nephroureterectomy (RNU) and excision of the ipsilateral bladder cuff. Both the NCCN and EAU Guidelines include neoadjuvant chemotherapy as a treatment option for select patients with UTUC; however, there are no strict guidelines. Much of the rationale for neoadjuvant chemotherapy is based on extrapolation from data from muscle-invasive bladder cancer, which has demonstrated a 5-year OS benefit of 5–8%. Retrospective studies evaluating the use of NACT in urothelial carcinoma have yielded pathologic objective response rates of 48% in UTUC cohorts. The randomized Phase III POUT study noted a DFS advantage with adjuvant platinum-based chemotherapy, compared with surveillance in UTUC, of 70% vs. 51% at 2 years. Though not the standard of care, multiple studies have explored the use of perioperative immunotherapy or chemoimmunotherapy in the management of invasive urothelial carcinoma. The PURE-02 study explored the use of neoadjuvant pembrolizumab in patients with high-risk UTUC. A small study of 10 patients, it showed no significant signals of activity with neoadjuvant pembrolizumab. Another Phase II study of neoadjuvant ipilimumab and nivolumab in cisplatin-ineligible UTUC yielded more promising findings, with 3/9 patients attaining a pathologic CR and the remaining six pathologically downstaged. The ABACUS trial found a 31% pathologic complete response rate amongst cisplatin-ineligible MIBC patients treated with neoadjuvant atezolizumab. The use of adjuvant immunotherapy has been explored over three phase III trials. The CheckMate-274 trial found a DFS benefit with the addition of one year of adjuvant nivolumab in patients with high-risk urothelial carcinoma. The IMvigor-010 study of adjuvant atezolizumab was a negative study. The AMBASSADOR trial of adjuvant pembrolizumab is pending results. With the FDA approval of erdafitinib in metastatic urothelial carcinoma, similar targets have been explored for use in perioperative use in invasive urothelial carcinoma, as with adjuvant infigratinib in the PROOF-302 trial. As the treatment paradigm for urothelial carcinoma evolves, further prospective studies are needed to expand the perioperative treatment landscape of UTUC.

## 1. Introduction

Urothelial cell carcinoma (UCC), also known as transitional cell carcinoma, is the predominant histological subtype of urinary tract cancer. It is the sixth most common tumor entity in developed countries [1]. The majority of these cases, approximately 90–95%, occur within the bladder. The remaining 5–10% originate in the upper urinary tract, which encompasses the renal pelvis, renal calyx, and the ureter [1]. This form is referred to as upper tract urothelial carcinoma (UTUC).

Although UTUC constitutes a minority of urothelial carcinomas, it is of particular interest due to its relatively aggressive biological behavior and higher mortality rate compared to other genitourinary tract malignancies. UTUC has a tendency to present at an advanced stage; at the time of diagnosis, around 60% of patients have invasive disease, and 30% present with metastatic disease [2]. This is in stark comparison to bladder cancer, where only 25% of patients present with invasive pathology [3]. Consequently, the overall 5-year disease specific survival is less favorable than lower tract UCC, ranging between 57 to 73% [4].

Global incidence of UTUC exhibits geographic variation. Western countries report a rate of 1–2 cases per 100,000 individuals annually, whereas in certain regions, such as Taiwan, UTUC comprises nearly a quarter of all urothelial carcinomas due to specific endemic factors [5]. Several risk factors for UTUC have been identified, including exposure to specific chemicals and drugs, tobacco smoking, prior pelvic radiation therapy, and inherited conditions such as Lynch syndrome. For instance, UTUC incidence is 2–3 times greater in individuals with a history of tobacco use, which is implicated in approximately 50% of cases in males and 33% in females [6]. Moreover, analgesic misuse and exposure to carcinogenic chemicals are associated with a fourfold and sixfold increase in risk, respectively. Furthermore, pathological risk factors significantly influence general outcomes in UTUC [6]. The European Association of Urology defines high-risk upper tract urothelial carcinoma as: high-grade cytology, high-grade-ureteroscopic biopsy, local invasion on CT, tumor size > 2 cm, multifocal disease, variant histology, and previous radical cystectomy for high-grade BC [7].

Despite advancements in diagnostic methodologies and therapeutic strategies, there has been limited improvement in the 5-year survival rate of UTUC over the preceding decades, underscoring the critical necessity for ongoing research. The advent of genomic sequencing techniques has facilitated the identification of a significant amount of genetic and epigenetic alterations within UTUC, thus providing a richer understanding of its pathogenesis and revealing novel potential therapeutic targets. The following sections will delve deeper into the intricacies of urothelial carcinoma of the upper urinary tract, shedding light on its diagnostic modalities, therapeutic approaches, and future directions in research. By doing so, we hope to provide a comprehensive understanding that may guide healthcare providers in delivering optimal care and contribute to the ongoing quest to improve outcomes for patients afflicted with UTUC.

## 2. Diagnosis

Upper tract urothelial carcinoma (UTUC) often elicits diagnostic consideration upon the manifestation of clinical symptoms such as hematuria and flank pain. Hematuria is particularly notable, present in an estimated 75% of UTUC cases (albeit non-exclusive to this malignancy, which consequently may contribute to diagnostic delay) [8]. Other common presenting symptoms are flank pain and presence of a lumbar mass occurring in 20–40% and 10–20% of cases, respectively [9]. The initial clinical suspicion is then followed by employment of imaging modalities which are integral to the detection and diagnosis of UTUC. Computed tomography urography (CTU) typically boasts high sensitivity and specificity, with respective rates ranging from 67–100% and 93–99% [8]. However, it is important to note that the sensitivity decreases to 89% for lesions less than 5 mm and 40% for lesions less than 3 mm, thereby presenting a potential limitation in diagnostic precision [10]. Alternative imaging modalities, such as magnetic resonance urography (MRU), offer similar rates of sensitivity and specificity to CTU.

Parallel to imaging, urine cytology adds another dimension to the diagnostic paradigm. Despite its high specificity of 94–98%, the sensitivity of cytology is quite variable, particularly for low-grade tumors, with sensitivity rates of 20–50%, increasing to 60–80% for high-grade tumors [11]. In an attempt to augment the sensitivity of UTUC detection, emerging research is directed towards the application of novel biomarkers, such as nuclear matrix protein 22 (NMP22) and fibroblast growth factor receptor 3 (FGFR3) mutation assays. Preliminary studies indicate encouraging results, with NMP22 demonstrating sensitivity and specificity rates up to 85% and 77% respectively [12].

The diagnostic paradigm for UTUC also incorporates the direct visualization and biopsy of suspected lesions via ureteroscopy. Biopsies in UTUC can be challenging due to difficulties accessing the upper urinary tract anatomy. Reported rates of nondiagnostic biopsies range from 10–20% [13]. Sampling error is a concern as UTUCs can be heterogeneous and solitary biopsies may miss higher grade components in up to 42% of cases [13]. There is also a small risk of tumor-seeding along the instrument tract during biopsy, estimated at <1% with proper technique [14]. Specimen interpretation is complicated by artifacts like crush and cautery effect, making it difficult to differentiate non-invasive from invasive disease in a significant number of samples. Furthermore, given the complexities of the procedure, the anatomy and the malignancy itself, upstaging from pT1 can occur in 61% of cases and upgrading from low to high grade can occur in 30% of cases [15]. There is no clear guideline on when a biopsy is absolutely needed versus proceeding directly to resection for suspected UTUC. Overall, issues with access, sampling, seeding risk, artifacts, and lack of consensus guidelines pose difficulties for biopsies in UTUC that require careful technique and interpretation.

The diagnosis of UTUC constitutes a complex, multifaceted process, encompassing aspects of clinical symptomatology, imaging, urine cytology, and endoscopic biopsy. Nonetheless, inherent limitations within each of these modalities underscore the imperative for continued advancement in diagnostic technologies and strategies. As research progresses, the future may see enhancements in imaging techniques, the introduction of novel biomarkers, and potentially novel diagnostic methodologies, all culminating in earlier and more accurate diagnoses of UTUC.

## 3. Treatment:

### 3.1. Perioperative Chemotherapy

Treatment of high-grade upper tract urothelial carcinoma (UTUC) with radical nephroureterectomy (RNU) and excision of the ipsilateral bladder cuff is standard for tumors of the renal pelvis. Endoscopic ablation and segmental ureterectomy can be considered for low-risk tumors, as is further discussed in subsequent chapters of this manuscript [2,16]. National Cancer Center Network (NCCN) and European Association of Urology (EAU) Guidelines do mention consideration of neoadjuvant chemotherapy in select patients with UTUC, though there are no strict guidelines [7]. The rationale for neoadjuvant chemotherapy (NAC), platinum-based, for UTUC is extrapolated from data in localized muscle-invasive bladder cancer (MIBC) to treat micro-metastatic disease, and downstage the tumor burden in those with optimal renal function [17]. Specifically, NAC prior to radical cystectomy in localized MIBC has shown a 5–8% improvement in OS at 5 years [18,19,20].

Unfortunately, the accurate staging of UTUC is much more challenging than in bladder cancer given the limitations and feasibility of biopsies [21,22,23]. Clinicians rely on radiologic imaging to clinically stage patients, though restaging is not often performed post-NAC and prior to RNU [24,25]. Despite these challenges, the multi-centric retrospective analysis by D’Andrea et al. showed similar outcomes of downstaging with use of NAC in both MIBC and UTUC [26,27]. In this study, a retrospective analysis was performed on 1830 patients treated with NAC, which was subsequently followed by radical cystectomy or RNU. Patients with metastatic disease were excluded from the trial. Results showed a pathological complete response in 19.2% of patients with urothelial carcinoma of the bladder (UCB) and 8.3% in patients with UTUC. A pathological objective response was seen in 40.3% of UCB patients and 48.2% in UTUC patients. In addition, Leow et al. also conducted a systematic review and meta-analysis that examined the efficacy of NAC and AC for non-metastatic UTUC. For NAC, pooled analysis of 14 studies (*n* = 811 patients) demonstrated an 11% pathologic complete response rate (defined as ≤ypT0N0M0) and 43% partial response rate (defined as ≤ypT1N0M0). Pathologic downstaging from the clinical tumor stage occurred in 33% across six studies. In comparative studies, NAC was associated with improved overall survival (OS) (hazard ratio [HR] 0.44, *p* < 0.001) and cancer-specific survival (CSS) (HR 0.38, *p* < 0.001) versus radical nephroureterectomy (RNU) alone. For AC, pooled analysis of 14 studies (*n* = 7983 patients) revealed an OS benefit (HR 0.77, *p* = 0.004), while 18 studies (*n* = 5659 patients) showed improved CSS (HR 0.79, *p* = 0.001) and 4 studies (*n* = 602 patients) demonstrated superior disease-free survival (HR 0.52, *p* < 0.001) with AC compared to RNU alone [28]. Overall, there have been many retrospective and prospective studies supporting peri-operative systemic therapy in the treatment of UTUC by benefiting improvement in OS and DSS, some of which we will outline in this review [29,30,31,32].

Neoadjuvant cisplatin-based therapy is the clinicians’ preference, rather than adjuvant platinum-based therapy, given the possible decline in renal function post-RNU which may render a patient ineligible for cisplatin-based therapy. One prospective study of neoadjuvant split-dose gemcitabine and cisplatin of 53 patients showed a CR of 19%, downstaging to ypT1 or less in 60% of patients, 2-year PFS of 76% [33]. Adjuvant cisplatin-based chemotherapy does also have a benefit in DFS as noted in the POUT study, a prospective, randomized phase III trial which showed DFS improvement at 2 years by 70% vs. 51% with the use of adjuvant platinum-based chemotherapy. Non-cisplatin-based therapies, including gemcitabine-based regimens, did not have an impact on mortality. The POUT trial arm of adjuvant carboplatin and gemcitabine for those with insufficient renal function noted that the DFS benefit at 3 years was upheld, though lacked improvement in OS [34]. Similarly, as in the treatment of MIBC, carboplatin-based regimens are not standard in either the neoadjuvant or adjuvant setting for patients who are cisplatin-eligible. However, carboplatin and gemcitabine can be considered adjuvantly in cisplatin-ineligible patients with high-risk upper tract disease.

Adibi et al. conducted a retrospective study between 2004–2017 of 126 patients with high-risk UTUC who were treated with NAC prior to RNU. NAC regimens did differ, as 62 received ddMVAC (methotrexate, vinblastine, Adriamycin, and cisplatin), 28 received cisplatin with or without gemcitabine, and 19 were treated with gemcitabine, paclitaxel, and doxorubicin. Seventeen patients received multiple different regimens or non-platinum-based therapy due to decreased renal function. Median OS was 107 months (95% CI 86–125), 14.3% achieved a pathologic complete response, while 60% were downstaged to ypT0-1N0. Estimated 5- and 10-year DSS rates were 89.8% (95% CI 0.836–0.965) and 80.6% (95% CI 0.691–0.94), respectively. Five- and 10-year metastasis free survival rates were 81% (95% CI 74–88.6) and 75.4% (95% CI 65.3–87), respectively, and 5- and 10-year OS were 73.7% (95% CI 65.3–83.1) and 35.9% (95% CI 23.9–54). Median time to recurrence was 15.5 months, with 24 metastatic recurrences documented, 50% to retroperitoneal, pelvic, of supraclavicular lymph nodes and 25% in the lung. This study supported the benefit of NAC prior to RNU with a durable 5-and 10-year OS and DSS [35]. Margulis et al. conducted a prospective multicenter phase II study consisting of 30 patients with high-grade UTUC receiving neoadjuvant accelerated MVAC (aMVAC) prior to nephroureterectomy. The pathologic complete response rate was 14% (4/29, 90% CI 4.9–28.8%). Overall, 62% achieved ≤pT1 at surgery. At a median 21 months follow-up, the 2-year recurrence-free and cancer-specific survival rates were 67% and 91%, respectively. Grade 3–4 toxicity occurred in 23% with aMVAC. While median creatinine clearance remained stable after chemotherapy (82 to 75 mL/min), it declined substantially to 48 mL/min after surgery, with 59% of patients becoming cisplatin-ineligible [29]. This study demonstrated neoadjuvant aMVAC appears safe and active for eligible patients with high-grade upper tract urothelial carcinoma, supporting further evaluation of this approach. Cisplatin ineligibility frequently develops after surgery, further highlighting the potential benefit of preoperative systemic therapy.

### 3.2. Perioperative Immunotherapy

There is limited data on the use of perioperative immunotherapy in the management of UTUC. The PURE-02 study was a feasibility study evaluating the use of three cycles of neoadjuvant pembrolizumab in patients with high-risk UTUC [36]. Despite the small sample size of 10 patients, there were no significant signals of activity with neoadjuvant pembrolizumab. Only one patient was characterized as a major responder, with a radiographic complete response to therapy. The remaining patients were defined as either nonresponders or with uncertain responses to therapy. A phase II study evaluated the use of neoadjuvant nivolumab and ipilimumab in patients with cisplatin-ineligible, high-grade UTUC [37]. The Stage I portion of the study enrolled nine patients, three of whom attained a pathologic CR (pCR); the remaining six patients were pathologically downstaged (<pT2pN0). Next-generation sequencing was performed on the pre-treatment tumor specimens. Interestingly, three patients were found to have germline variants in mismatch repair genes; one attained a pCR and the other two ypTaN0. The ABACUS trial was a single-arm, phase II study evaluating two cycles of neoadjuvant atezolizumab prior to cystectomy in 95 cisplatin-ineligible muscle-invasive bladder cancer patients. At a median follow-up of 25 months, the 2-year disease-free and overall survival rates were 68% (95% CI 58–76%) and 77% (95% CI 68–85%), respectively. In the 31% of patients achieving a pathologic complete response, the 2-year disease-free survival rate was 85% (95% CI 65–94%). High baseline stromal CD8+ T cells and negative baseline circulating tumor DNA status correlated with improved relapse-free survival, while post-treatment fibroblast activation protein positivity was associated with worse outcomes. Serial circulating tumor DNA analysis demonstrated conversion to negative status after neoadjuvant therapy in some baseline-positive patients and was highly prognostic for relapse when positive post-cystectomy [38]. In summary, atezolizumab showed promising preliminary efficacy in patients with MIBC. Further research is warranted to confirm these findings and determine if similar efficacy and safety of neoadjuvant atezolizumab can be reproduced in patients with UTUC prior to radical nephroureterectomy.

Gao et al. further evaluated the efficacy of PD-L1 plus CTLA-4 blockade in the neo-adjuvant setting. In this open-label, single-arm pilot study, 28 cisplatin-ineligible patients with high-risk muscle-invasive urothelial carcinoma received neoadjuvant durvalumab plus tremelimumab every 4 weeks for two doses. The pathologic complete response (pCR) rate was 38% (9/24 patients, 95% CI 19–59%) among those completing cystectomy. In 12 patients with T3/T4 disease, the pCR rate was 42% (5/12 patients). The overall downstaging rate to ≤pT1N0 was 58% (14/24 patients, 95% CI 36–77%). At a median follow-up of 19 months, the 1-year overall survival rate was 89% (95% CI 70–96%) and 1-year relapse-free survival rate was 83% (95% CI 61–93%). Grade ≥ 3 treatment-related adverse events occurred in 21% (6/28 patients) of patients. High baseline tertiary lymphoid structure density was significantly associated with improved survival. [39] In summary, neoadjuvant durvalumab plus tremelimumab showed encouraging antitumor activity and manageable toxicity in high-risk cisplatin-ineligible muscle-invasive bladder cancer. Two ongoing trials are evaluating the use of neoadjuvant durvalumab combined with chemotherapy for patients with high-risk UTUC [40,41].

Several Phase III studies have explored the use of adjuvant immunotherapy in patients with high-risk muscle-invasive urothelial carcinoma. The CheckMate-274 trial, which randomized patients to receive one-year of adjuvant nivolumab or placebo, found a disease-free survival benefit with the addition of adjuvant nivolumab; overall survival results are not mature. Of the 709 patients in the intention-to-treat population, 149 had upper tract disease [42]. On subgroup analysis, there was no benefit of the addition of nivolumab for upper tract disease (HR [renal pelvis] 1.23, 95% CI 0.67–2.23; HR [ureter] 1.56 95% CI 0.70–3.48); however, the study was not powered to specifically evaluate this. The IMvigor-010 study evaluated the use of adjuvant atezolizumab in patients with locally advanced or metastatic UTUC who had previously received platinum-based chemotherapy. This was a negative trial with no disease-free survival benefit in the intention-to-treat population (HR 0.89, 95% CI 0.74–1.08; *p*= 0.24) [43]. On the horizon is the AMBASSADOR trial, which is a phase III randomized, double-blind, placebo-controlled clinical trial evaluating the use of adjuvant pembrolizumab after nephroureterectomy in patients with high-risk UTUC. The trial aimed to enroll 360 patients who underwent radical nephroureterectomy for high-risk, non-metastatic UTUC. Patients were randomized 1:1 to receive either adjuvant pembrolizumab every 3 weeks or placebo for up to eighteen cycles. The primary endpoint is overall survival and disease-free survival. Key eligibility criteria include: high-grade UTUC (either high-grade papillary cancer or invasive urothelial carcinoma), pT2-T4 or pTany with positive lymph nodes, and no neoadjuvant chemotherapy. The results from this trial have the potential to establish adjuvant pembrolizumab as a new standard of care for high-risk UTUC patients after nephroureterectomy. The trial is expected to be completed in 2025 [44].

## 4. Future Directions

In recent years, advancements in the genomic understanding of UTUC have delineated the potential therapeutic promise of the Fibroblast Growth Factor Receptor (FGFR) pathway. Comprising a group of receptor tyrosine kinases, FGFR is instrumental in the regulation of critical cellular processes, including proliferation, differentiation, and survival. The dysregulation of FGFR signaling pathways, predominantly due to gene mutations or fusions, is implicated in the tumorigenesis of a wide range of cancers, UTUC included. The prevalence of FGFR mutations or fusions in UTUC is relatively high, with genetic alterations involving FGFR reported in an estimated 20% of patients with advanced urothelial cell cancer [45]. This revelation has sparked substantial interest in the development of FGFR-targeted therapies, resulting in several clinical trials examining the potential benefits of FGFR inhibitors, such as erdafitinib and infigratinib, for UTUC.

Erdafitinib, an FGFR1-4 inhibitor, has been tested in the clinical setting for patients with UTUC and other urothelial carcinomas. It was granted accelerated approval by the FDA based primarily on the results of a multicenter, open-label, single-arm study conducted by Loriot et al. In this open-label phase 2 trial, 99 patients with metastatic or unresectable urothelial carcinoma harboring FGFR alterations received the pan-FGFR inhibitor erdafitinib continuously at 8 mg or 9 mg daily doses. The confirmed objective response rate was 40% (95% CI 31–50%). Among FGFR mutation patients, the response rate was 49%. Median progression-free survival was 5.5 months and median overall survival was 13.8 months. The 12-month overall survival rate was 55%. Grade ≥ 3 treatment-related adverse events occurred in 46% of patients, most commonly hyponatremia, stomatitis and asthenia [46]. Erdafitinib showed promising antitumor activity and manageable toxicity in this patient population who had progressed on prior chemotherapy and/or immunotherapy.

Infigratinib, a selective FGFR1-3 inhibitor, has also shown promise in FGFR-altered urothelial cancer, including UTUC. Lyou et al. conducted an open-label multicenter phase 1b study, where 13 patients received the FGFR1-3 inhibitor infigratinib early-line before platinum chemotherapy for metastatic urothelial carcinoma, while 54 received it after ≥1 prior therapies. The confirmed objective response rate was 31% (4/13 patients, 95% CI 9.1–61.4%) with early-line and 24% (13/54 patients, 95% CI 13.5–37.6%) with later-line infigratinib. Disease control rates were 46% (6/13 patients, 95% CI 19.2–74.9%) and 69% (37/54 patients, 95% CI 54.4–80.5%) in the early-line and salvage settings, respectively. Median progression-free survival was 12.0 months versus 5.6 months, and median overall survival was 13.8 months versus 12.9 months in the early-line compared to later-line groups [47]. Infigratinib demonstrated clinically meaningful antitumor activity regardless of treatment line in metastatic urothelial carcinoma, supporting further evaluation across different settings.

Notwithstanding the encouraging preliminary results of FGFR-targeted therapies, multiple challenges endure. These include the emergence of resistance to FGFR inhibitors, the management of therapy-associated side effects, and the necessity for reliable biomarkers to guide patient selection. Moreover, the definitive impact of FGFR inhibitors on overall survival remains under investigation. These studies represent important strides in the treatment of UTUC and other urothelial carcinomas. They underscore the importance of genomic profiling in urothelial cancer to identify patients who might benefit from these targeted treatments. As further studies are conducted, the role of FGFR inhibitors in the treatment paradigm of UTUC is likely to become better defined.

In addition to immunotherapy and chemotherapy, there is significant interest in the role of radiation therapy in the adjuvant space. A systematic review by Iwata et al. evaluated the role of adjuvant radiotherapy (ART) after surgery for bladder cancer and UTUC. For bladder cancer, the review included three randomized controlled trials comprising 456 patients and 11 retrospective studies comprising 7571 patients [48]. Some studies found ART improved recurrence-free survival (5-year rates of 49% vs. 25% in one RCT) and local recurrence-free survival (5-year rates of 87% vs. 50% in one RCT), but most studies found no statistically significant impact on metastasis-free or overall survival [48]. For UTUC, 14 retrospective studies comprising 6047 patients were included. Most studies did not find a survival advantage for ART, except two studies that showed improved overall survival in locally advanced UTUC (29.9 vs. 11.4 months in one study) [48]. Toxicity from ART is decreasing with improved radiotherapy techniques, with recent studies showing lower rates of severe gastrointestinal toxicity and bowel obstruction compared to older studies [48]. The quality and quantity of data on ART in bladder cancer and UTUC was found to be limited. The combination of ART and chemotherapy may be beneficial for locally advanced tumors. The authors concluded there is currently no clear evidence for the survival benefit of ART after surgery for bladder cancer or UTUC, and future efforts should focus on multimodal therapy with ART plus chemotherapy or immunotherapy.

## 5. Conclusions

Further prospective, randomized clinical trials of peri-operative chemo- and immunotherapy in the treatment of UTUC are needed to answer efficacy questions and establish a new standard of care.

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
