# Peer review of "Overview, Diagnosis, and Perioperative Systemic Therapy of Upper Tract Urothelial Carcinoma"

_cancers, 2023, doi:10.3390/cancers15194813_

Round 1
Reviewer 1 Report
Manuscript entitled "Perioperative Systemic Therapy of Upper Tract Urothelial Carcinoma"
1. The definition of localized disease, high-risk disease ... etc., should be mentioned clearly in tables.
2. The biomarkers for the selection of (neo)adjuvant therapy should be listed for further clarification.
3. The role of traditional chemotherapy and radiation therapy should be mentioned.
4. The strategies/methods for the surveillance of recurrence should be mentioned.
Acceptable
Author Response
Reviewer 1
Definition of localized disease, high-risk disease, should be mentioned clearly in the tables (DONE)
The following text was added to Figure 1:
Low-Risk disease - Low-risk UTUC refers to localized, low-grade tumors with small volume, papillary architecture, limited depth of invasion (non-invasive or lamina propria only), and lack of aggressive features like carcinoma in situ, lymphovascular invasion, lymph node metastases, or radiographic evidence of advanced disease.
High-risk UTUC refers to tumors with aggressive features like high-grade disease (G3), carcinoma in situ (CIS), large size (>2cm), infiltrative architecture, multifocality, advanced local extent on imaging, muscularis propria invasion or beyond (≥pT2), lymphovascular invasion (LVI), lymph node involvement or metastases (pN1-3 or M1), or high volume even if low stage/grade.
Biomarkers for the selection of (neo)adjuvant therapy should be listed for further clarification (DONE)
There are currently no standard of care biomarkers being used for evaluation for perioperative therapy. However, we do discuss the possible use of FGFR2/3 alterations on trial (PROOF-302) in the “Future Direction” section.
Role of traditional chemotherapy and radiation therapy should be mentioned (DONE)
Discussed radiation therapy, starting at line 301. Role of chemotherapy is discussed in detail throughout section 3
Strategies/methods for surveillance of recurrence should be mentioned (DONE)
This is not the focus of the paper. A critique from Reviewer 2 was that we strayed too far off from the topic.
Reviewer 2 Report
I appreciate the effort the authors put into the study; however, after careful evaluation, I do feel that the paper cannot be considered for publication in its current form.
I would like to highlight two major points that influenced this decision:
Title-Content Mismatch: The title of your paper does not accurately correspond to the information presented within the manuscript. The content appears to deviate significantly from the expectations set by the title (e.g. long elaboration on risk factors and diagnosis). A well-defined and representative title is crucial for setting the reader's expectations and ensuring alignment with the content.
Lack of Methodology: The methodology employed in your study is not adequately explained in the manuscript. A clear description of the methods used is essential for the readers and reviewers to assess the validity and reliability of your findings. Without a proper methodology section, the overall rigor of the study is compromised. Search queries, databases used, and other restrictions for the studies inclusion into the review should be mentioned.
Please find several other comments below:
1. In line 50 you mentioned that UTUC originates from ureter or renal pelvis. You should have also mentioned renal calyx or in general pelvicalyceal system.
2. The introduction is quite long and many unnecessary details about risk factors are mentioned. Risk factors for UTUC development were not the topic of the paper.
3. In lines 71-73 authors mention different criteria for high-risk UTUC and state that these are associated with worse outcomes. This sentence lacks comparison. Worse than who? Secondly, these factors are preoperative ones, which was not clearly mentioned.
4. The second point of the paper describes the diagnostic scheme for UTUC. This is not directly related to the perioperative chemotherapy mentioned in the titile and in the aim of the paper.
5. The sentence in lines 130-131 lacks accuracy. The treatment depends on the preoperative risk and the location of the tumour. There are other surgical approaches than RNU which is standard for high-risk cases originating from the renal pelvis and proximal ureter. Segmental ureterectomy and endoscopic ablation techniques were not mentioned.
6. In lines 134-135 it is mentioned that evidence for NACH is extrapolated from bladder cancer to UTUC setting. This is not quite accurate. There is a meta-analysis and at best level 2 evidence for perioperative CHTX in UTUC.
The paper lacks important citation of the meta-anlysis about perioperative chemotherapy for UTUC published two years ago. DOI: 10.1016/j.eururo.2020.07.003
Above systematic review and meta-analysis are perhaps the best available evidence for the use of chemotherapy in neo- or adjuvant setting.
7. Please correct the sentence “Similarly as in the treatment of MIBC, carboplatin-based regimens are not standard in 164 either the neoadjuvant nor adjuvant setting.” This is not the standard in patients eligible for cisplatin, but in patients with impaired renal function, this is the preferred choice.
8. Abascus trial is mentioned to prove the efficacy of neoadjuvant atezolizumab prior to cystectomy. If you mention this please underline that the data is not available for UTUC. Otherwise it can be confusing for the reader and can lead to wrong conclusions.
9. I think that subsections should be created to divide the information about neoadjuvant chemotherapy, adjuvant chemotherapy, neoadjuvant immunotherapy, adjuvant immunotherapy. This would be helpful for the readers. Now the text is hard to follow.
10. Table 1 lacks a report on the treatment efficacy (e.g. DFS/OS) in each trial.
Author Response
Reviewer 2
Title-Content Mismatch – (DONE)
Changed title to ”Overview, Diagnosis, Treatment, and Perioperative Systemic Therapy of UTUC”
Lack of Methodology – This paper is NOT a systematic review and meta-analysis. Rather, it is an overview of the diagnosis and perioperative treatment options for upper tract disease.
1. Line 50 – add renal calyx or in general pelvicalyceal system:
This was added.
2. Length of intro and diagnosis section too long
Changed title to include overview and diagnosis
3. Lines 71-73 – different criteria for high-risk UTUC and state that these are associated with worse outcomes. This sentence lacks comparison. Worse than who? Secondly, these factors are preoperative ones, which was not clearly mentioned.
This sentence was edited as follows: The European Association of Urology defines high risk upper tract urothelial carcinoma as: high-grade cytology, high-grade-ureteroscopic biopsy, local invasion on CT, tumor size > 2cm, multifocal disease, variant histology, and previous radical cystectomy for high grade BC.7
4. The second point of the paper describes the diagnostic scheme for UTUC. This is not directly related to the perioperative chemotherapy mentioned in the titile and in the aim of the paper. Please see updated title.
5. The sentence in lines 130-131 lacks accuracy. The treatment depends on the preoperative risk and the location of the tumour. There are other surgical approaches than RNU which is standard for high-risk cases originating from the renal pelvis and proximal ureter. Segmental ureterectomy and endoscopic ablation techniques were not mentioned. ???AUA guildeines
This was briefly addressed in the paper (see lines 131-134) and will be addressed in detail in subsequent chapters in the special issue. The focus of this paper is on systemic therapy, not surgical management.
6. In lines 134-135 it is mentioned that evidence for NACH is extrapolated from bladder cancer to UTUC setting. This is not quite accurate. There is a meta-analysis and at best level 2 evidence for perioperative CHTX in UTUC.
The paper lacks important citation of the meta-anlysis about perioperative chemotherapy for UTUC published two years ago. DOI: 10.1016/j.eururo.2020.07.003
Above systematic review and meta-analysis are perhaps the best available evidence for the use of chemotherapy in neo- or adjuvant setting.
The following was added, per your recommendation:
In addition, Leow et al. also conducted a systematic review and meta-analysis that examined the efficacy of NAC and AC for non-metastatic UTUC. For NAC, pooled analysis of 14 studies (n=811 patients) demonstrated a 11% pathologic complete response rate (defined as ≤ypT0N0M0) and 43% partial response rate (defined as ≤ypT1N0M0). Pathologic downstaging from clinical tumor stage occurred in 33% across 6 studies. In comparative studies, NAC was associated with improved overall survival (OS) (hazard ratio [HR] 0.44, p<0.001) and cancer-specific survival (CSS) (HR 0.38, p<0.001) versus radical nephroureterectomy (RNU) alone. For AC, pooled analysis of 14 studies (n=7983 patients) revealed an OS benefit (HR 0.77, p=0.004), while 18 studies (n=5659 patients) showed improved CSS (HR 0.79, p=0.001) and 4 studies (n=602 patients) demonstrated superior disease-free survival (HR 0.52, p<0.001) with AC compared to RNU alone.
7. Please correct the sentence “Similarly as in the treatment of MIBC, carboplatin-based regimens are not standard in 164 either the neoadjuvant nor adjuvant setting.” This is not the standard in patients eligible for cisplatin, but in patients with impaired renal function, this is the preferred choice.
Correction:
Similarly as in the treatment of MIBC, carboplatin-based regimens are not standard in either the neoadjuvant nor adjuvant setting for patients who are cisplatin-eligible. However, carboplatin and gemcitabine can be considered adjuvantly in cisplatin-ineligible patients with high-risk upper tract disease.
8. Abascus trial is mentioned to prove the efficacy of neoadjuvant atezolizumab prior to cystectomy. If you mention this please underline that the data is not available for UTUC. Otherwise it can be confusing for the reader and can lead to wrong conclusions.
The following was added, per your recommendation:
In summary, atezolizumab showed promising preliminary efficacy in patients with MIBC. Further research is warranted to confirm these findings and determine if similar efficacy and safety of neoadjuvant atezolizumab can be reproduced in patients with UTUC prior to radical nephroureterectomy.
9. I think that subsections should be created to divide the information about neoadjuvant chemotherapy, adjuvant chemotherapy, neoadjuvant immunotherapy, adjuvant immunotherapy. This would be helpful for the readers. Now the text is hard to follow.
We have divided section 3 (“Treatment”) into perioperative chemotherapy (Section 3a) and perioperative immunotherapy (Section 3b).
10. Table 1 lacks a report on the treatment efficacy (e.g. DFS/OS) in each trial.
As suggested, added treatment efficacy column to Table 1
Reviewer 3 Report
The authors reviewed the diagnosis and perioperative medical treatments for upper tract urothelial carcinoma. This review came together very well, so it contains very useful information for urologists and seems to be extremely valuable. Further improvement is expected by the following points.
1) The AUA also recently published guidelines for UTUC. Please also cite these guidelines to enhance the value of this review.
Author Response
1) The AUA also recently published guidelines for UTUC. Please also cite these guidelines to enhance the value of this review.
As the AUA guidelines focus on surgical management, we have not included it here. Surgical management will be discussed in subsequent chapters in the special issue.
Reviewer 4 Report
well performed review.
Author Response
No suggestions from the reviewer
Round 2
Reviewer 1 Report
The revision is acceptable for publication.
Acceptable.
Reviewer 2 Report
No further comments. Thank you